# Experimental Study on Long-Term Mechanical Properties and Durability of Waste Glass Added to OPC Concrete

**DOI:** 10.3390/ma16175921

**Published:** 2023-08-29

**Authors:** Jichao Zhu, Xinyu Meng, Baoyuan Wang, Qianhao Tong

**Affiliations:** 1School of Civil Engineering, Dalian Jiaotong University, Dalian 116028, China; zjc@djtu.edu.cn (J.Z.);; 2State Key Laboratory of Bridge Engineering Structural Dynamics, China Merchants Chongqing, Communications Technology Research & Design Institute Co., Ltd., Chongqing 400015, China; 3School of Mechanics & Civil Engineering, China University of Mining and Technology, Xuzhou 221116, China; 4Chery Automobile Co., Ltd., Wuhu 241000, China

**Keywords:** waste glass powder, compressive strength, data fitting, chloride ion permeability coefficient, freeze–thaw cycling

## Abstract

This study aims to achieve the sustainable utilization of waste glass resources through an investigation into the influence of three types of admixtures, namely waste glass powder (WGP) (G), waste glass powder–slag (G-S), and waste glass powder–fly ash (G-F), on the mechanical properties and durability performance of waste glass concrete. The experimental results demonstrate that the exclusive use of WGP as an admixture led to the relatively poor early compressive strength of the concrete, which decreased with an increase in dosage. However, at medium to long curing ages, the strength of the waste glass concrete could equal or even surpass that of ordinary concrete. When dual admixtures were employed, the G-S group exhibited higher compressive strength compared to the G-F group. Specifically, within the G-S group, a glass powder dosage of 15% yielded higher compressive strength, and after 180 days, the dual admixture groups exhibited greater strength than ordinary concrete (G0); the compressive strength of the tG1S1 group was 44.57 MPa, and that of the G0 group was 40.07 MPa. The chloride ion diffusion coefficient showed a varying trend with an increase in WGP dosage, initially decreasing and then increasing. The concrete’s resistance to erosion was maximized when the glass powder dosage reached 30%. As the WGP dosage increased, the overall frost resistance decreased. For a total dosage of 30%, the optimal glass powder dosage in both G-S and G-F groups was found to be 15%.

## 1. Introduction

Concrete, one of the most extensively utilized construction materials on a global scale, requires the use of ordinary Portland cement (OPC) in its production process, which emits a significant amount of carbon dioxide (CO_2_) and other greenhouse gases. Consequently, the cement industry has been recognized as one of the foremost and least ignorable causes of global warming [1]. The reduction of carbon emissions from cement production has emerged as a focal area of research for numerous scholars in recent years [2,3]. Noviani et al. [4] argue that the widespread adoption of more environmentally friendly composite silicate cement should be pursued. Some researchers have explored the incorporation of construction waste such as waste glass into concrete [5,6] and found that it not only effectively recycles the increasing volume of waste glass produced annually but also enhances the durability properties of concrete to some extent. Furthermore, unlike other construction waste, waste glass cannot naturally decompose, and its proper disposal is challenging [7], giving rise to the issue of how to efficiently utilize waste glass.

In the initial stage, many scholars investigated the use of waste glass as aggregate in concrete and found through experiments [8,9,10,11,12] that directly incorporating waste glass particles as coarse aggregate can lead to an alkali-aggregate reaction, which is influenced by the particle size. When the particle size is larger than 1 mm, the alkali-aggregate reaction becomes more significant as the particle size increases. With further research, some researchers have adjusted the proportion of waste glass to other components in concrete, seeking the optimal combination. It was found that when waste glass particles act as fine aggregates, they can improve the workability of the mixture [11,13]. By substituting 25% of cement with waste glass powder (WGP), blocks were prepared. Mechanical and durability performance tests were conducted on individual blocks, as well as on composite blocks, to compare their mechanical properties. The results showed that the block group with 10% WGP substitution exhibited similar performance to the control group [14]. Furthermore, other researchers have investigated the substitution of WGP at different proportions as a replacement for quartz-aggregate powder, examining its effects on the dosage of polymer concrete resin, concrete workability, as well as its impact on compressive strength and flexural strength [15]. WGP obtained from crushing containers and demolishing buildings was utilized to produce glass powder blended cement as an additive in concrete [16]. The volcanic ash activity of glass powder and the performance of glass powder cement were evaluated. Additionally, the influence of using crushed WGP and ground granulated blast furnace slag as partial substitutes for cement on the mechanical properties and durability of concrete was studied [17]. Moreover, the chemical erosion of sulfate solution on reinforced concrete tanks with the addition of WGP was investigated [18]. The sustainable production of green infrastructure permeable concrete was achieved by utilizing waste materials such as fly ash and WGP as partial substitutes for cement [19]. Different types of WGP were employed as substitutes for fine aggregate in high-performance concrete (HPC) [20]. Compared to the complete replacement of fine aggregate, a replacement rate between 20% and 40% exhibits inferior mechanical and durability properties in the short-term age, but it surpasses the benchmark concrete in the medium to long-term curing period [21,22,23,24]. It can be observed that substituting waste glass particles for both coarse and fine aggregates is not entirely reliable. To address the issues of alkali-aggregate reaction and early strength deficiency when incorporating waste glass into concrete, many scholars have shifted their research focus to grinding waste glass particles and replacing cement or using them as supplementary cementitious materials, namely mineral admixtures, in concrete. Through experimental analysis, some researchers demonstrated that finely ground waste glass powder, when partially replacing cement [25,26], exhibits an activity index exceeding 70% to 80% of that of fly ash materials under natural curing conditions, possessing properties comparable to cement and mineral admixtures. When the substitution rate of WGP for cement does not exceed 25%, its impact on concrete can be neglected. Among them, when the volume ratio of WGP replacing cement is less than 20% and the particle size is less than 135 microns, the working performance is better, the compressive strength of short age (28 d) is increased by 25% [27], and the mechanical properties over the long term (180 d) are significantly improved [28,29,30]. It can function as a cementitious material [31,32], possessing higher erosion resistance and durability [33]. However, it still reduces early strength in concrete [34]. To improve early strength and balance the mechanical performance distribution during concrete curing, WGP is used in dual admixture with mineral admixtures. Tran et al. [35] demonstrated through two types of experiments that WGP and slag powder exhibit improved workability and mechanical properties. The 20% substitution rate of glass powder demonstrates better compressive and flexural strength, effectively enhancing early strength and durability, as well as contributing to low carbon environmental sustainability. By leveraging the inherent alkali-rich nature and volcanic ash characteristics of waste glass powder, a value-added treatment approach was employed to enhance its properties. This treatment process aimed to improve the corrosion resistance of waste glass powder, thereby bolstering the durability of high-performance recycled aggregate concrete (RAC) [36].

In summary, the application of waste glass in concrete has achieved some results by reference value, but the research is mostly focused on the mechanical properties of short-age concrete, and few studies systematically consider the long-term mechanical properties of glass powder concrete (glass powder as an admixture), the deterioration rule under freeze–thaw action, and the ability to resist chlorine salt erosion. Because of this, this paper considers the influence of environmental factors in the northern coastal area of China and investigates the anti-chlorine erosion performance of glass powder (G) single-mixed as well as glass powder-mineral double-mixed (G-F/G-S) concrete, the mass loss after freeze–thaw cycle, the relative modulus of elasticity, and the compressive strength of different compounding ratios at different ages, to reveal the long-term mechanical properties and durability of glass powder concrete, and provide a theoretical basis and a basis for the popularization and application of glass powder concrete. The study reveals the long-term mechanical properties and durability of glass powder concrete and provides a theoretical basis and data support for the popularization and application of glass powder concrete.

## 2. Materials and Methods

### 2.1. Materials

The experiment used ordinary Portland cement (PO42.5R) produced by Xiaoyetian Cement Factory in Dalian City, Liaoning Province, China, with a compressive strength of 53 MPa and a flexural strength of 9.1 MPa at 28 days. The waste glass used was ordinary transparent plate glass recycled from domestic use by Hangzhou Gaoke Composite Materials Co., Ltd. in Hangzhou, Zhejiang Province, China, with a particle size of 40 μm. It was ground into powder as shown in Figure 1, and its main chemical composition is shown in Table 1.

The slag powder and fly ash were both obtained from Ruian Longze Materials Co., Ltd. in Ruian, Zhejiang Province, China, with grade 1 fly ash and S95 grade slag powder. The details of their basic properties and chemical compositions can be found in Table 2 and Table 3, respectively.

The fine aggregate used was local river sand in Dalian, and the coarse aggregate was crushed stone with a particle size of 5–25 mm. Water was ordinary tap water from Dalian, and a standard high-performance polycarboxylate superplasticizer was used as the water-reducing agent.

### 2.2. Methods

The present experiment focused on the study of single-admixture and dual-admixture concrete, with waste glass powder, fly ash, and slag powder as the research parameters. For single admixture concrete, WGP was substituted for cement in volume ratios of 10%, 20%, 30%, 40%, and 50%. For dual admixture concrete, the total admixture content was kept at 30%, with a water-to-binder ratio of 0.46. The basic mix proportions were determined based on the experimental methods specified in JCJ55-2011 “Standard for Mix Proportion Design of Ordinary Concrete” [37], and adjusted according to the actual experimental conditions. Eleven mix proportions were designed as shown in Table 4, where G, F, and S represent waste glass powder, fly ash, and slag powder, respectively.

According to the specifications provided in JTG E30-2005 “Test Regulations for Cement and Concrete in Highway Engineering” [38] regarding the relationship between specimen dimensions and the maximum aggregate size for different test purposes, when the maximum aggregate size is less than 26.5 mm, the mechanical test specimens have dimensions of 100 × 100 × 100 mm, the electrical flux test specimens have dimensions of Φ100 × 50 mm, and the frost resistance test specimens have dimensions of 100 × 100 × 400 mm. Table 5 shows the number of specimens prepared for each test and the curing ages are presented.

This article conducted a total of one mechanical performance test and two durability tests, including compressive strength test, electrical flux test, and freeze-thaw cycling test.

The compressive strength test of the specimens was conducted using a DYE3000 pressure testing machine of Shengke Test Instrument Co., Ltd., Dalian, Liaoning Province, China, with the loading rate of 0.5–0.8 MPa/s. The test was stopped when the peak stress decreased to 40%. The electrical flux test was carried out using a DTL-6T chloride ion flux measuring instrument and a concrete vacuum saturation device of Dalian Shengke Test Instrument Co., Ltd. The chloride ion permeability of the tested specimens was calculated based on the flux measurement, and the chloride ion diffusion coefficient was also determined. The freeze–thaw cycling test was performed using a KD13–28V concrete rapid freeze–thaw testing machine of Shengke Test Instrument Co., Ltd. Dalian, Liaoning Province, China. During the cycling process, the lowest temperature at the center of the specimen was −18 °C, while the highest temperature was 5 °C. The solution used was pure water.

## 3. Results and Discussion

### 3.1. Mechanical Performance

The study incorporated various curing ages for conducting mechanical performance tests. Compressive strength tests were carried out on concrete specimens with different mix proportions at different curing ages. The experimental results revealed that when the applied load reached 85% of the ultimate load, cracks appeared on the surface of the specimen. As the load continued to increase, the specimen appeared transversely uplifted, the surface fell off, and finally collapsed, which is considered ductile destruction. Failure patterns of specimens with different admixtures and different dosage are basically the same, as illustrated in Figure 2.

Figure 3 depicts the compressive strength development of concrete specimens with single-admixture WGP at different curing ages.

It can be observed that the compressive strength of the single-admixture WGP concrete shows a positive correlation with the curing age. At the same curing age, as the replacement ratio of WGP increases, the compressive strength generally exhibits a downward trend. Further investigation reveals that with the increase in curing age, the difference in strength between the GX group and the G0 group becomes smaller. Compared to the G0 group, the G3 group experienced a strength reduction of 25.4% at 3 days and 8.01% at 180 days. This phenomenon is attributed to the pozzolanic effect of the waste glass powder, which becomes prominent in long-term curing, promoting the formation of C-S-H gel and enhancing the rate of strength development in the WGP concrete. Among all of the groups, G1 exhibited the highest growth rate, surpassing the strength of the G0 group by 4.1% at 180 days.

The influence of different glass powder dosages on the compressive strength of dual-admixture concrete at various curing ages is illustrated in Figure 4. Overall, the compressive strength exhibits a positive correlation with the curing age.

During the early curing ages (3–14 d), the compressive strength of the glass powder concrete is relatively slow. The G0 group increased by 9.44 MPa, the G-F group exhibited an average compressive strength growth of 8.85 MPa, and the G-S group had an average compressive strength growth of 10.04 MPa. In the medium- to long-term curing ages (28–180 d), the increase in compressive strength of the dual-admixture concrete becomes more significant. The G-F group showed an average increase in compressive strength of 8.12 MPa, the G-S group had an average increase of 8.07 MPa, and the G0 group experienced a growth of 6.68 MPa. It is worth noting that the compressive strength of both the G1F1 and G1S1 groups exceeded that of the other groups after 56 days. On the whole, the G-S concrete exhibited higher final strength than the G-F concrete. This indicates that the pozzolanic effect brought about by glass powder and slag powder has greater advantages in the long-term curing period compared to fly ash. This phenomenon may be attributed to the inert characteristics of glass powder, and the slag powder can better enhance the activity of glass powder by dissolving a large amount of silica. This promotes the reaction between glass powder and the hydration product Ca(OH)_2_, resulting in the formation of C-S-H gel and further enhancing the compressive strength of the concrete in the later stages [30,33,39,40].

Compressive strength, as one of the important indicators in concrete design, is a key parameter that characterizes its mechanical performance. Based on the observed mechanism of the variation in compressive strength with the curing age, it is evident that the compressive strength of concrete varies at different curing ages. For predicting strength after 28 days, a time-dependent model can be employed for fitting and prediction purposes:(1)$$f_{cu,t}=\alpha f_{cu,28}\mathrm{lg}\left(\frac{t}{28}\right)+f_{cu,28}$$
where t is the age of the specimen in days; $f_{cu,t}$ refers to the compressive strength of the cube at the curing age t; and α is a constant coefficient.

Transforming Equation (1) leads to Equation (2):(2)$$y=\alpha f_{cu,28}x$$
where y=$\left(f_{cu,t}-f_{cu,28}\right)/f_{cu,28}$ (MPa); x=$\mathrm{lg}\left(\frac{t}{28}\right)$.

Utilizing Equation (2), the compressive strength of the blended concrete cubes with the admixture was fitted against the curing age, resulting in Figure 5.

The horizontal y-axis represents a function variable with the curing age t, while the vertical axis represents the growth rate of compressive strength at t days of age compared with 28 days. The fitting results are presented in Table 6.

Table 6 presents the fitting coefficients comparing the dual-blended concrete with single-blended and ordinary concrete. From Figure 5, it can be observed that the larger the slope, the faster the compressive strength of the concrete increases. For single-blended concrete, as the amount of WGP increases, the slope significantly increases. Compared with Figure 3, WGP can enhance the rate of compressive strength growth in concrete, but it does not significantly improve early strength. For dual-blended concrete, the G1F1 and G1S1 groups exhibit higher slopes and faster growth in compressive strength. Considering the G1S1 group shows a good linear correlation between curing age and growth rate, with a fitting coefficient of 0.99, it can be concluded that slag powder can stabilize the activity of waste glass powder, thereby improving compressive strength. This model can reasonably simulate the compressive strength of WGP single/dual-blended concrete to a certain extent, thereby providing useful references for engineering design.

### 3.2. Durability Performance

#### 3.2.1. Chloride Ion Attack

The treatment of specimen electrical flux was conducted based on Equation (3), and the electrical flux values were converted according to Equation (4) depending on the size of the specimens. The conversion of concrete electrical flux values to chloride ion diffusion coefficient was carried out using an empirical Formula (5) proposed by Feng et al. [41,42,43] from Tsinghua University. The results of data processing are presented in Table 7, and the data from the table are plotted in Figure 6.
(3)$$Q=900\left(I_{0}+2I_{30}+2I_{90}+\cdots +2I_{300}+2I_{330}+2I_{360}\right)$$
where Q is the total electrical flux (C) passing through the specimen within 6 h; $I_{0}$ is the initial current (A); $I_{t}$ is the current value passing through the specimen at time *t* (A).
(4)$$Q_{S}=Q_{X}\times {\left(\frac{95}{x}\right)}^{2}$$
where $Q_{S}$ is the electric flux through a specimen with a diameter of 95 mm (C); $Q_{X}$ is the actual electric flux through the specimen used in the experiment (C); x is the actual diameter of the cylindrical specimen in the experiment (mm).
(5)y=2.57765+0.00492x
where y is the chloride ion diffusion coefficient (×10^−2^ cm^2^/s); x is electric flux passed by concrete specimens during translation (C).

Based on the above data, it is evident that the concrete exhibits the strongest resistance against chloride ion penetration when the chloride ion diffusion coefficient is within the range of 3.07–7.50 (×10^−9^ cm^2^/s). When glass powder is solely added, the overall trend of concrete resistance against chloride ion penetration shows an initial increase followed by a decrease. The electrical flux is minimum when the glass powder dosage is 30%, indicating the highest resistance against chloride ion erosion and the lowest chloride ion diffusion coefficient of 7 × 10^−9^ cm^2^/s. In the case of the simultaneous addition of glass powder and mineral admixture, the higher the proportion of glass powder, the lower the electrical flux. When the glass powder proportion is at 20% and 15%, the concrete exhibits even stronger resistance against chloride ion penetration. In conclusion, glass powder has the most significant impact on the resistance of concrete against chloride ion penetration, with an optimal dosage of 30% in sole addition, and 20% glass powder proportion in simultaneous addition.

#### 3.2.2. Freeze–Thaw Cycling

Mass loss rate

Based on the change pattern of the rate of mass variation in different mineral-admixed concrete with a total dosage of 30% as shown in Figure 7, it is observed that although some specimens show an increase in mass during the early stage of freeze–thaw cycles, the overall mass loss rate tends to increase.

During the freeze–thaw cycling process, cracks propagate from the surface toward the interior. When surface cracking occurs, the pore width expands, allowing more free water to penetrate the structure, promoting the hydration reaction to a certain extent, and resulting in an increase in the mass of the concrete specimens. However, as the number of cycles increases, the rate of surface detachment accelerates, leading to an overall phenomenon of initial mass increase followed by a decrease.

Comparing the various groups of specimens, it was found that the mass loss rate of the specimens in Group G3 increased to 5% after 100 freeze–thaw cycles, reaching the failure condition. In contrast, the specimens in the dual-admixture group did not exhibit a mass change rate exceeding 5% even after 100 cycles, indicating that the addition of glass powder as a sole admixture has an adverse effect on the frost resistance of concrete due to its smooth surface and poor bondability. In the dual-admixture concrete, as the proportion of glass powder increased, the mass loss rate showed a trend of an initial decrease followed by an increase. Throughout the entire freeze–thaw cycle, the mass loss rate of the specimens with 15% glass powder dosage did not reach the failure condition, while the other groups experienced failure. This indicates that under the condition of a total dosage of 30%, the optimal frost resistance is achieved when the proportion of glass powder is 50%.

2.Relative loss of dynamic modulus

Figure 8 represents the influence of dual admixture on the relative dynamic modulus of elasticity of concrete.

Except for Group G3, the frost resistance of the concrete in the other groups showed minimal deviation from the G0 concrete. With an increase in the number of freeze–thaw cycles, all groups gradually deteriorated, exhibiting a good linear relationship. When comparing the specimens in each group, Group G3, due to the incorporation of a higher proportion of glass powder, exhibited lower initial reactivity, resulting in a significant amount of dehydration particles within the concrete system. This led to weaker structural bonding forces, increased internal porosity, and greater vulnerability to damage. As a result, the relative dynamic modulus of elasticity of the Group G3 specimens was already below 85% after 25 freeze–thaw cycles, significantly lower than the dual-admixture groups as well as the G0 group. After 125 cycles, the relative dynamic modulus of elasticity decreased to below 60%.

In contrast, the dual-admixture groups fully utilized the pozzolanic effect, reducing the influence of crack development on the specimens and creating a more compact internal structure in the concrete. Among them, the G-S group exhibited a higher relative dynamic modulus of elasticity. Within the dual-admixture groups, the specimens with a glass powder dosage of 15% experienced the lowest decrease in the relative dynamic modulus of elasticity, indicating the best frost resistance. None of the dual-admixture groups experienced failure after the completion of the freeze–thaw cycle.

## 4. Conclusions

Based on the comparative analysis of compressive strength, resistance to chloride ion erosion, and frost resistance at different ages of WGP concrete, waste glass powder–fly ash concrete, and waste glass powder–slag powder concrete, the following main conclusions can be drawn:The sole addition of WGP led to a reduction in the early-age compressive strength of the concrete. The volcanic ash effect mainly occurred after 28 d. In the case of dual admixture, concrete will enter a period of rapid strength growth in advance. When the total dosage was 30%, a 1:1 ratio of WGP to mineral admixture yielded a higher compressive strength, with the G−S group specimens exhibiting superior compressive strength compared to the G−F group.Predictions of compressive strength variation with age were conducted using a time-dependent model. It was found that as age increases, compressive strength shows an overall increasing trend. The greater the dosage of glass powder, the larger the slope, and the faster the rate of strength growth. The average coefficient of determination (R^2^) for the regression equation of the time-dependent model was 0.92, indicating that the model can be used to predict compressive strength values at different ages.Both sole and dual admixture concrete showed improved resistance to chloride ion erosion when the total dosage was 30%. When the glass powder was solely added at a dosage of 30%, the chloride ion diffusion coefficient was the lowest. In the case of dual admixture, the higher the dosage of glass powder, the stronger the resistance to chloride ion erosion, showing an overall positive correlation.In both sole and dual admixture concrete, the overall frost resistance tended to decrease with an increase in the dosage of glass powder. When the total dosage was 30%, both the G−F and G−S groups exhibited improved frost-resistance characteristics. The optimal frost resistance was achieved when the glass powder and mineral admixture each accounted for 15%.

## Figures and Tables

**Figure 1 materials-16-05921-f001:**
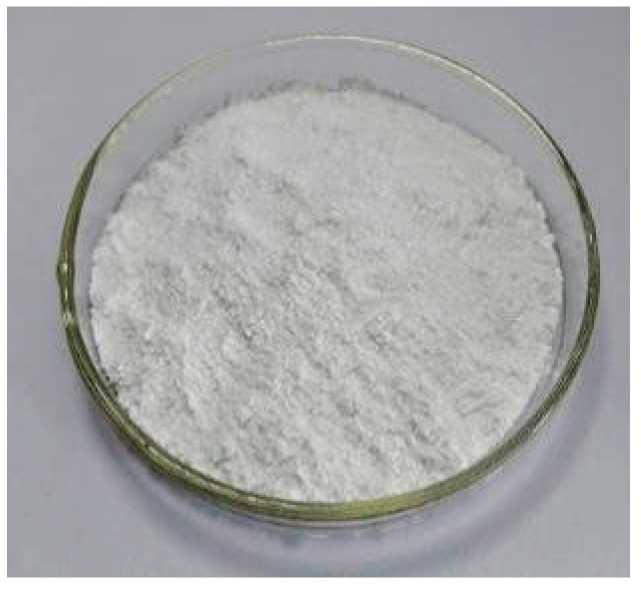
Waste glass powder.

**Figure 2 materials-16-05921-f002:**
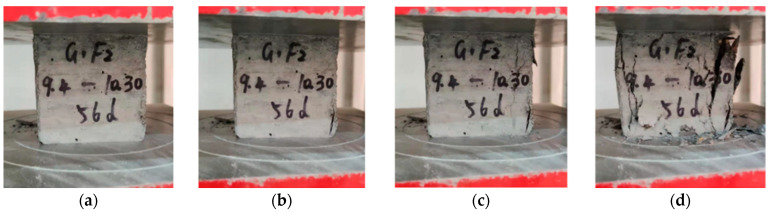
Failure morphology of a specimen, (**a**) under compression, (**b**) crack initiation, (**c**) crack propagation, (**d**) specimen failure.

**Figure 3 materials-16-05921-f003:**
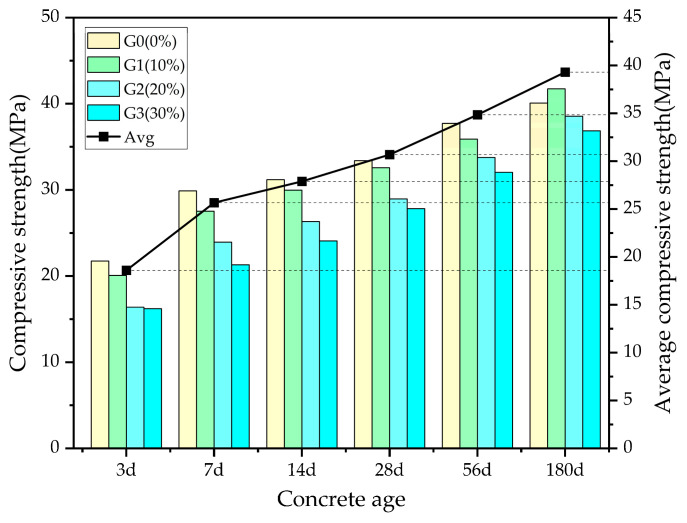
Compressive strength of unmixed concrete.

**Figure 4 materials-16-05921-f004:**
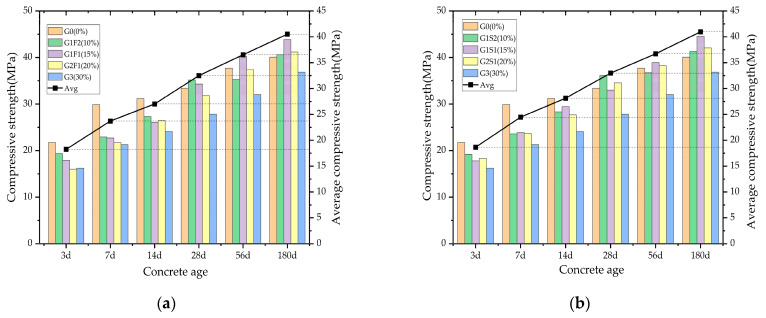
Compressive strength of double-mixed concrete: (**a**) G-F, (**b**) G-S.

**Figure 5 materials-16-05921-f005:**
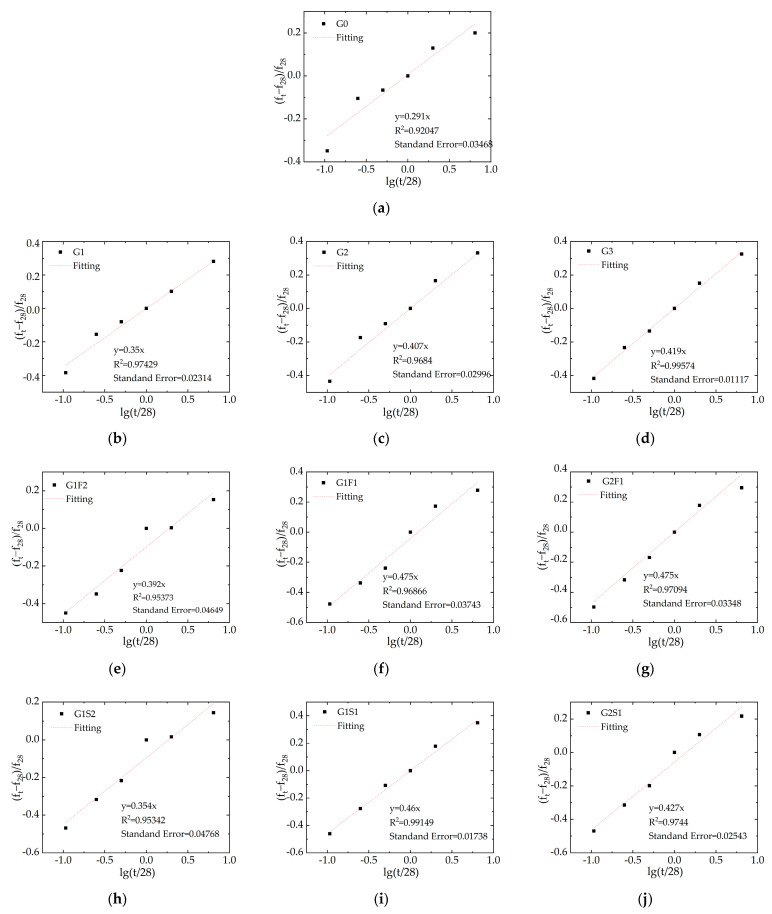
Fitting of concrete compressive strength, (**a**) G0, (**b**) G1, (**c**) G2, (**d**) G3, (**e**) G1F2, (**f**) G1F1, (**g**) G2F1, (**h**) G1S2, (**i**) G1S1, (**j**) G2S1.

**Figure 6 materials-16-05921-f006:**
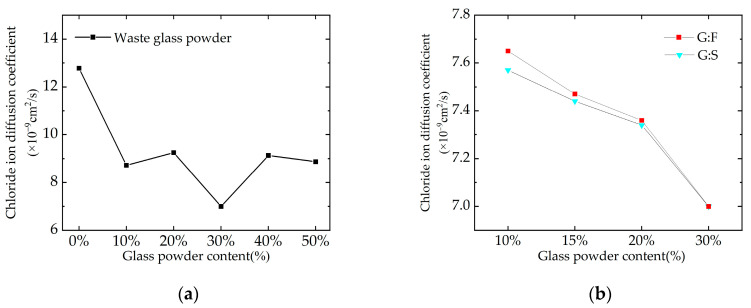
Variation trend of chloride ion diffusion coefficient in concrete: (**a**) single-blended concrete, (**b**) dual-blended concrete.

**Figure 7 materials-16-05921-f007:**
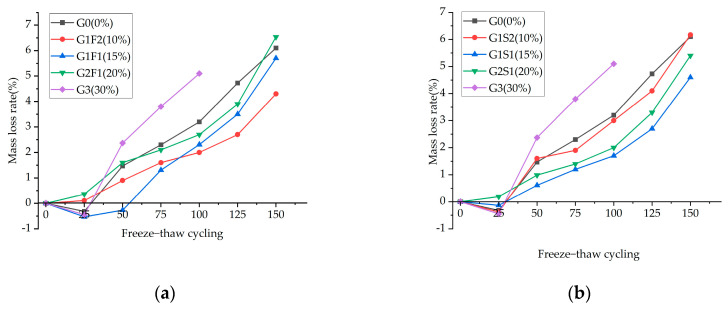
Mass loss rates for (**a**) G−F, (**b**) G−S.

**Figure 8 materials-16-05921-f008:**
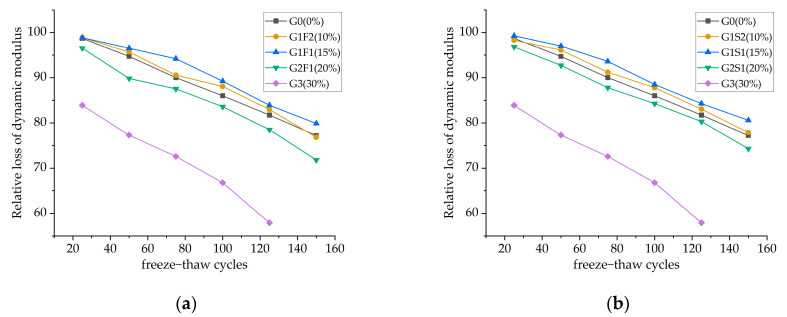
Relative loss of dynamic modulus for (**a**) G−F, (**b**) G−S.

**Table 1 materials-16-05921-t001:** Main chemical components of waste glass powder.

Composition	SiO_2_	Na_2_O	CaO	Al_2_O_3_	K_2_O	Fe_2_O_3_	MgO	SO_3_	TiO_2_
Unit (wt.%)	67.79	13.89	11.37	3.51	0.87	0.53	0.62	0.80	0.15

**Table 2 materials-16-05921-t002:** Chemical composition and properties of fly ash.

**Performance**	**Specific Surface Area (m^2^/kg)**	**Flow Ratio (%)**	**Density (g/cm^3^)**	**Moisture Content (%)**	**Activity Index (%)**		
					**7 d**		**28 d**
	429	98	3.1	0.45	84.2		98.5
**Composition**	**MgO**	**SiO_2_**	**CaO**	**Al_2_O_3_**	**Fe_2_O_3_**	**SO_3_**	**Loss on ignition (%)**
Unit (wt.%)	6.01	34.5	34	17.7	1.03	1.64	0.84

**Table 3 materials-16-05921-t003:** Test values of chemical composition and properties of slag powder.

**Performance**	**Specific Surface Area (m^2^/kg)**	**Flow Ratio (%)**	**Density (g/cm^3^)**	**Moisture Content (%)**	**Activity Index (%)**		
					**7 d**		**28 d**
	430	97	3.1	0.43	84.1		98.3
**Composition**	**CaO**	**SiO_2_**	**Al_2_O_3_**	**SO_3_**	**Fe_2_O_3_**	**MgO**	**Loss on ignition (%)**
Unit (wt.%)	34	34.2	17.6	1.62	1.01	6.21	0.87

**Table 4 materials-16-05921-t004:** Concrete mix ratio.

Code	Waste Glass Powder	Fly Ash	Slag Powder	Cement	Water	Sand	Stone
G0	0	0	0	355	163	625	1185
G1	35.5 (10%)	0	0	319.5	163	625	1185
G2	71 (20%)	0	0	284	163	625	1185
G3	106.5 (30%)	0	0	248.5	163	625	1185
G4	142 (40%)	0	0	142	163	625	1185
G5	177.5 (50%)	0	0	177.5	163	625	1185
G1F2	35.5 (10%)	71 (20%)	0	248.5	163	625	1185
G1F1	53.25 (15%)	53.25 (15%)	0	248.5	163	625	1185
G2F1	71 (20%)	35.5 (10%)	0	248.5	163	625	1185
G1S2	35.5 (10%)	0	71 (20%)	248.5	163	625	1185
G1S1	53.25 (15%)	0	53.25 (15%)	248.5	163	625	1185
G2S1	71 (20%)	0	35.5 (10%)	248.5	163	625	1185

**Table 5 materials-16-05921-t005:** Statistical table of specimens (pieces).

Experimental	Compressive Strength	Electrical Flux	Freeze-Thaw Cycling
Age (d)	3, 7, 14, 28, 56, 180	56	60
Groups	60	8	10
Quantities	6	3	3
Subtotal	360	24	30
Grand Total	414		

**Table 6 materials-16-05921-t006:** Analysis results of compressive test data.

Code	Fitting Model	R^2^
G3	$f_{cu,t}=0.419f_{cu,28}\mathrm{lg}\left(\frac{t}{28}\right)+f_{cu,28}$	0.996
G1F1	$f_{cu,t}=0.475f_{cu,28}\mathrm{lg}\left(\frac{t}{28}\right)+f_{cu,28}$	0.969
G1S1	$f_{cu,t}=0.46f_{cu,28}\mathrm{lg}\left(\frac{t}{28}\right)+f_{cu,28}$	0.991

**Table 7 materials-16-05921-t007:** Test results of chloride ion penetration in concrete.

Code	Electric Flux (C)	Chloride Ion Penetration Resistance Grade	Chloride Ion Diffusion Coefficient (×10^−9^ cm^2^/s)
G0	2073.65	middle	12.78
G1	1248.45	low	8.72
G2	1356.17	low	9.25
G3	899.79	extremely low	7.00
G4	1331.78	low	9.13
G5	1278.93	low	8.87
G1S2	1014.71	low	7.57
G1S1	987.34	extremely low	7.44
G2S1	968.68	extremely low	7.34
G1F2	1030.97	low	7.65
G1F1	994.38	extremely low	7.47
G2F1	972.02	extremely low	7.36

## Data Availability

Not applicable.

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
