# Peer review of "Experimental Study on Long-Term Mechanical Properties and Durability of Waste Glass Added to OPC Concrete"

_materials, 2023, doi:10.3390/ma16175921_

Round 1
Reviewer 1 Report
1. In section 2.1, can authors discuss the crushing process and screening process of the recycled waste glass and particle sizes?
2. Table 6 and fig 5 shows the R2 higher than 0.95 after curing for 28days, how about 3-14 days curing and 28-180 days curing?
3. Please double check the average compressive strength of G group specimens listed in Figure 3.
4. Please double check the average values in Figure 4(a).
5. Can authors explain the reason why the compressive strength of G-F and F-S group (10% glass) decrease after curing 28-56days ?
Reviewer 2 Report
Overall the paper is well written, following the course of actions determined in the begining.
As I'm not familiar with Chinese standards (namely JTG E30-2005 which seems to be succeeded in 2020 by other standard) are all the methods used for the sample testing specified in this standard/conducted accorting to this standard? If not please elaborate more on the test methodology.
Why were mixes G4 and G5 used only for chloride ion penetration test and no toher data are shown for these mixes?
Please use the same range for main- and side- y-axis in figures 3 and 4. Why have figures 6 a and b have reversed x-axis? Please revise figure 7 to have the same axis intersection in both parts.
In the discussion of mechanical properties, the authors discuss growth rate, which is not properly defined in the text, so it is unsure if it is obtained by realting the strength of same mixture specimens in different ages or comparing the different mixtures at same age. Please revise all the discussion to make it more clear. What is the reason behind plotting the average strenth at each age (as I understand it it is the average of strengths of all different mixes at the same age, which may show some time-releated dependencies however, it cannot show any differences in the behaviour of different mixes).
Reviewer 3 Report
The manuscript presents a topic which has already seen a lot of published work. The results are innovative only because the long term durability has been considered. Authors should be more precise in the title and in the abstract when they describe their investigation.
Title
I propose the following title: "Experimental Study on Long-Term Mechanical Properties and Durability of Waste Glass added to OPC Concrete"
Abstract
First sentence: "..... three types of waste glass powder ......". The differences are starting aluminosilicate source in terms of admixtures: G, G-F. G-S. Please correct the first sentence.
Can you add the amount of WGP added in the different formulations?
Could you add numerical results for your experimental data mentioned in the abstract?
Introduction
The Introduction should be integrated with comments on the following published papers:
https://doi.org/10.1016/j.conbuildmat.2020.119115
https://doi.org/10.1016/j.conbuildmat.2021.122400
Experimental Investigation on Strength of Glass Powder Replacement by Cement in Concrete with Different Dosages (IJSTE/ Volume 2 / Issue 08 / 013)
https://doi.org/10.3390/pr11040975
https://doi.org/10.21307/acee-2020-030
The novelty and originality of this paper should be stressed in the Introduction after having inserted these new references.
Experimental
Be more precise when you indicate the provenance of your raw materials:
Line 111: Dalian 110 Xiaoye Cement Plant,??? indicate province and country.
Line 113: Hangzhou Gaoke Composite Materials Co., Ltd., same comment as above.
Line 112: "ordinary transparent glass recycled from daily life" do you mean container glass? Flat glass? borosilicate glass? WEEE glass? In our daily life we use a lot of different glasses.
Line 114: ",,,chemical composition is shown in Table 1." Please comment about the analytical technique used to evaluate the chemical composition of your raw materials.
Table 1 reports: Unit(vol.%). Please change to wt%=weight% since the composition you are reporting is much closer to wt% than vol%.
Table 2: same comments for the chemical composition in wt%. What is: Turnover Ratio(vol.%)???? Are you sure that the Moisture Content is measured in vol.%????? What is; Activity Index(vol.%)????? What is the Firing Vector????
Table 3: What is Fineness in vol%?? What is the Firing Vector???? What is: Water Demand Ratio in vol%???
Line 130: ".........was substituted for cement in mass ratios of 10%, 20%, 30%, 40%, and 50%." Mass ratio is not expressed in %! Be more consistent and use the proper unit.
Results and Discussion: OK
Conclusions: OK
Authors should try to use shorter and clearer sentences, mainly in the abstract.
Reviewer 4 Report
Need for the current study and the research gap has to be clearly addressed.
Slag source as well as its chemical composition has to be verified and detailed.
What failure pattern was observed for all the mixes under direct compression? Need to be elaborated.
From what tool, the modeling has been carried and how the equations were arrived?
How many samples were cast for each mix so as to address the error stats?
Images on the specimens exposed to various chemical exposures to be presented.
Does the results correspond to chloride ion refers to the corrosion characteristics?
XRD patterns of the specimens exposed to various chemical attacks need to be presented in order to justify the components responsible for the change in behaviour.
Add references to justify the variation in the behvaiour of the specimens.
Conclusions should not be a discussion of the test results. It should be short and crisp.
Round 2
Reviewer 3 Report
The manuscript has been improved. All corrections have been addressed carefully. Nevertheless, there is still an error to be amended:
Line 38: "135 microns" change microns in "micrometres" or use the appropriate symbol.
Reviewer 4 Report
Comments were taken care